# Associations between Dietary Patterns and Cardiometabolic Risk Factors—A Longitudinal Analysis among High-Risk Individuals for Diabetes in Kerala, India

**DOI:** 10.3390/nu14030662

**Published:** 2022-02-04

**Authors:** Yingting Cao, Quan Huynh, Nitin Kapoor, Panniyammakal Jeemon, Gabrielli Thais de Mello, Brian Oldenburg, Kavumpurathu Raman Thankappan, Thirunavukkarasu Sathish

**Affiliations:** 1Melbourne School of Population and Global Health, University of Melbourne, Melbourne 3053, Australia; nitin.endocrine@gmail.com (N.K.); brian.oldenburg2@baker.edu.au (B.O.); speaktosat@gmail.com (T.S.); 2Implementation Science Lab, Baker Heart and Diabetes Institute, Melbourne 3004, Australia; 3Imaging Research, Baker Heart and Diabetes Institute, Melbourne 3004, Australia; quan.huynh@baker.edu.au; 4Department of Endocrinology, Diabetes and Metabolism, Christian Medical College, Vellore 632004, India; 5Sree Chitra Tirunal Institute for Medical Sciences and Technology, Trivandrum 695011, India; jeemon@sctimst.ac.in (P.J.); kr.thankappan@gmail.com (K.R.T.); 6Research Centre for Physical Activity and Health (NuPAF), Federal University of Santa Catarina, Florianópolis 88040-900, Brazil; gabi.tmello@hotmail.com; 7School of Psychology and Public Health, La Trobe University, Melbourne 3086, Australia; 8Department of Public Health and Community Medicine, Central University of Kerala, Kasaragod 671316, India; 9Population Health Research Institute (PHRI), McMaster University, Hamilton, ON L8L 2X2, Canada

**Keywords:** principal component analysis, elevated Hb1Ac, central obesity, multi-level mixed effects modeling

## Abstract

The association between dietary patterns and cardiometabolic risk factors is not well understood among adults in India, particularly among those at high risk for diabetes. For this study, we analyzed the data of 1007 participants (age 30–60 years) from baseline and year one and two follow-ups from the Kerala Diabetes Prevention Program using multi-level mixed effects modelling. Dietary intake was measured using a quantitative food frequency questionnaire, and dietary patterns were identified using principal component analysis. Two dietary patterns were identified: a “snack-fruit” pattern (highly loaded with fats and oils, snacks, and fruits) and a “rice-meat-refined wheat” pattern (highly loaded with meat, rice, and refined wheat). The “snack-fruit” pattern was associated with increased triglycerides (mg/dL) (β = 6.76, 95% CI 2.63–10.89), while the “rice-meat-refined wheat” pattern was associated with elevated Hb1Ac (percentage) (β = 0.04, 95% CI 0.01, 0.07) and central obesity (OR 1.16, 95% CI 1.01, 1.34). These findings may help inform designing dietary interventions for the prevention of diabetes and improving cardiometabolic risk factors in high-diabetes-risk individuals in the Indian setting.

## 1. Introduction

India is estimated to have the second largest number of people with diabetes globally (74 million), and this is projected to increase to 125 million (nearly 70% increase) by 2045 [1]. While the prevalence of diabetes in India is highest in the more economically and epidemiologically advanced states, an increase in less advanced states is also occurring at a significantly rapid pace, resulting in a large national diabetes burden [2]. In addition, obesity, particularly central obesity, is a major health problem in India, with a prevalence ranging from 16.9 to 36.3% in different states [3]. Together with the unique “atherogenic dyslipidemia profile” and a “South Asian phenotype”, Indians are at a higher risk of developing cardiometabolic diseases than many other ethnic groups [4].

Among many modifiable factors that contribute to the development of cardiometabolic diseases, diet plays a critical role, and a healthy diet is associated with improved cardiometabolic health [5,6]. In recent years, there has been a shift from focusing on individual foods or their nutrients’ consumption to assessing dietary patterns, which considers the complex combinations of foods and nutrients that are highly intercorrelated [7]. Dietary patterns reflect one’s long-term eating habits, which cannot be measured directly. Most used statistical methods to measure dietary patterns are factor analysis, using principal component analysis [7]. Research to understand dietary patterns in India has been very limited. A systematic review published in 2016 identified only eight studies on dietary patterns conducted in India [8]. In total, 11 models with 41 separate dietary patterns were identified [8], and 29 out of 41 dietary patterns were predominantly vegetarian food groups, composing of fruits, vegetables, and pulses and cereals (mostly rice), with added dairies, meet and eggs. 

The association between dietary patterns and cardiometabolic risk are well established from studies conducted in Western countries. In general, these studies show that a “healthy” dietary pattern that is rich in vegetables and fruits is inversely associated with cardiometabolic risk, whereas a “unhealthy” dietary pattern that is rich in foods such as red meat, processed food, and fried food is positively associated with cardiometabolic risk [9,10,11]. However, Indian dietary patterns are more complex than a simplified “healthy” or “unhealthy” dietary pattern seen in Western countries. The “unhealthy” dietary pattern identified in Western populations that is characterized by red meat, sweetened beverages, and processed food are not typically consumed in the Indian population. Instead, diets with high-carbohydrate (refined), high-fat and high-salt (fried carbohydrate, pulses as snacks), together with some fruits, are commonly consumed in India, and this doesn’t seem to be easily detected as “unhealthy” as in the Western diet. However, together they can have higher cardiometabolic risk [12]. For example, a common dietary pattern is an “animal-food” pattern (characterized by meat, poultry, fish, and eggs) in the Indian population, which is different from the Western “unhealthy” pattern that is usually characterized by red meat, processed meat, and fried foods. This Indian “animal-food” pattern was found to be positively associated with obesity and central obesity [13]. Likewise, the Indian “cereals-savory foods” pattern (characterized by grains, rice, and condiments) that is not usually observed in Western populations was found to be protective of obesity and central obesity [13].

Even within India, dietary patterns differ between Northern and Southern regions, where Northern patterns are generally characterized by animal products, fried snacks, and sweets, whereas Southern patterns compose of increased consumption of fruits, vegetables, pulses, and rice [12]. However, all these studies were cross-sectional, and prospective studies are needed to understand the long-term effect of dietary patterns on cardiometabolic risk factors in India. More importantly, there has been a “nutrition transition” in India in the past decades that has resulted in changes in diet (e.g., reduced intake of coarse cereals, fruits, and vegetables, and increased intake of unhealthy fats, salt, and animal foods). This has been observed in parallel to the rapid development of economy and the epidemic of cardiometabolic diseases [14]. Therefore, more research is needed to better identify the link between diet and cardiometabolic diseases risk in India.

Given the huge diversity of diet across regions and highly prevalent cardiometabolic risk in India, dietary patterns identified in specific regions or states may be hard to compare. Therefore, understanding diet in places with higher cardiometabolic risk populations might have to be prioritized. Kerala state in South India has become a model for population-based studies and research into cardiometabolic diseases, as the state that is in an advanced epidemiological transition and has a higher burden of several cardiometabolic risk factors than most other states in the country [15]. Our team has established a population-based cohort in Kerala in 2013 to implement and evaluate a community-based, cluster randomized controlled trial of a peer-led lifestyle intervention program in high-risk populations for diabetes, the Kerala Diabetes Prevention Program (K-DPP) [16]. The two-year follow-up of this trial has provided the opportunity to study the longitudinal association between dietary patterns and cardiometabolic risk factors. In this study, we aimed to: (1) identify dietary patterns among individuals at high risk for diabetes in Kerala; and (2) to examine the longitudinal association between dietary patterns and cardiometabolic risk factors over 2 years.

## 2. Materials and Methods

### 2.1. Study Participants

The detailed information on the K-DPP study design and participant screening and recruitment have been previously published [16,17]. In brief, participants were randomly selected from 60 polling areas (electoral divisions) from the Neyyattinkara taluk (sub-district) in Trivandrum district of Kerala. A total of 3421 potential participants were screened for eligibility and people with the following conditions were excluded: (1) those with a prior history of diabetes; (2) those who had other major chronic illnesses; (3) those who were taking medications affecting glucose tolerance (e.g., corticosteroids); (4) those who were illiterate in the local language; and (5) pregnant women. The potentially eligible participants (*n* = 2586) underwent a two-step screening procedure, which included the Indian Diabetes Risk Score (IDRS) and a 2 h 75 g oral glucose tolerance test (OGTT) [18].

After excluding those with an IDRS score of <60 (*n* = 1057) and those who were not willing to have an OGTT (*n* = 320), 1,209 participants underwent the OGTT. A total of 1007 participants without diabetes at baseline were included in the K-DPP trial (intervention group 500, control group 507). For this analysis, we included K-DPP participants (*n* = 1007) at baseline, with a follow-up at year 1 (n = 982) and a follow-up at year 2 (*n* = 964) (Figure 1).

### 2.2. Cardiometabolic Factors

Anthropometric measurements were performed using standard protocols in clinics organized in community neighborhoods [19], including height, weight, waist, and hip circumferences, as well as body composition. In brief, height was measured with a portable Seca stadiometer (model 213) to the nearest 0.1 cm with the subject standing straight with feet together, head in Frankfort plane, and arms hanging freely. Weight was measured using an electronic scale to the nearest 0.1 kg (TANITA body composition model SC330) while barefoot and wearing light clothing. Waist circumference was measured midway between the lowest rib and the iliac crest, using a Seca measuring tape (model 201) in accordance with the WHO STEPS protocol [20]. Blood pressure (BP) was recorded three times using the Omron automatic BP monitor (model IA2) with an interval of at least 3 min between the readings. The average of the second and third BP readings was used in the current analysis. Standard protocols were followed for the collection of fasting glucose, OGTT, HbA1c, and lipids [16]. Blood samples were centrifuged for 30 min at the clinic and transported to a laboratory accredited by the National Accreditation Board for Laboratories (NABL) [21] for processing.

Body mass index (BMI) was computed as weight (kg)/height (m²) and was categorized according to the Asia Pacific guidelines [22]: normal weight: BMI < 23 kg/m^2^; overweight: 23 kg/m^2^ to ≤ BMI < 25 kg/m^2^; and obese: BMI ≥ 25 kg/m^2^. Metabolic syndrome was defined as per the International Diabetes Federation guidelines [23]: central obesity (waist circumference ≥90 cm for males and ≥80 cm for females) + any two or more of the following: (1) hypertriglyceridemia (triglycerides ≥ 150 mg/dL (1.7 mmol/L); (2) low HDL cholesterol (<40 mg/dL (1 mmol/L) for males and <50 mg/dL (1.3 mmol/L) for females); (3) elevated blood pressure (systolic blood pressure ≥ 130 mmHg and /or diastolic blood pressure ≥ 85 mmHg or drug treatment for hypertension); and 4) elevated fasting blood glucose ≥ 100 mg/dL (5.6 mmol/L). Prediabetes was defined according to the ADA: 5.6 mmol/L ≤ Fasting glucose ≤ 6.9 mmol/L or 7.8 mmol/L ≤ 2 h glucose ≤ 11.0 mmol/L, or 5.7% ≤ HBA1c ≤ 6.4% [24]. Diabetes was diagnosed according to the American Diabetes Association (ADA) criteria (fasting plasma glucose (FPG) ≥ 7.0 mmol/L and/or 2 h postprandial plasma glucose (PPG) ≥ 11.1 mmol/L) on the OGTT [24].

### 2.3. Dietary Measures and Dietary Patterns

Dietary intake was measured using a quantitative food frequency questionnaire (FFQ), which asks frequency of usual dietary intake (number of portions consumed on a daily, weekly, monthly, yearly/never basis) over the last 12 months. The FFQ was adapted from a previous study, the PROLIFE study, conducted in Kerala (see Appendix A) [25]. The FFQ administered in the current study included 53 food items (extra food items were also obtained by asking “other food items” under each food group).

All the food items were classified into 22 groups, which were adapted from food groups from previous Indian studies conducted in the same region and by consulting with local nutritionists [12,26]. The intake of each food (in grams) was computed to daily equivalents for analyses. Dietary patterns across three timepoints (baseline, year 1, and year 2) were identified by principal component analysis (data reduction techniques) with estimated daily intake in grams. Varimax rotation was used to improve interpretability and to minimize the correlation between the factors. The final number of components (dietary patterns) was determined by (1) eigenvalue > 1.25, (2) scree plot, and (3) interpretability of the factors [27]. Each food group under each component was assigned a factor loading. Factor loadings represent simple correlations between the food groups and the component or the pattern. In other words, higher absolute values indicate a higher association between the food group and the component or the pattern, meaning that the food group shares more variance with the component or the pattern. Factor loadings were graphically presented. The dietary components (patterns) were named subjectively according to the characteristics of the structure of the factor loadings for each of the patterns.

A factor score for each pattern at each timepoint was assigned for each participant by summing the total grams of the 22 food groups (standardized), which were weighted by their factor loadings. Similarly, a higher score indicates a greater association with the specific pattern. Factor scores were used for analysis. In this study, a mean cumulative score (the sum of scores over three time points divided by three) for each participant was used for analysis, as it has been shown to better reflect long-term diet and may reduce measurement error [28]. Only participants with complete dietary intake data for all the three timepoints were included in the current analysis (*n* = 958).

### 2.4. Key Confounding Variables

Interviewer-administered questionnaires were used to collect data on sociodemographic factors (e.g., age, sex, marriage status, education, and occupation) and behavioral factors (e.g., tobacco use, alcohol use, and physical activity). Tobacco use was assessed by asking the question “Did you use any of the following tobacco products (smoking: cigarettes, bidis, cigars and hookah; smokeless: snuff, betel with tobacco, khaini, and gutka) in the last 30 days?” Alcohol consumption was assessed by asking the question “Did you consume an alcoholic drink (such as beer, wine, whiskey, toddy) in the last 30 days?’ Self-reported levels of physical activity were measured using the Global Physical Activity Questionnaire [29].

### 2.5. Statistical Analysis

Basic characteristics at baseline were summarized according to dietary patterns. The mean cumulative score (SD) of each dietary pattern was compared by categories of demographic factors. ANOVA was used to compare differences in the mean cumulative scores between groups. Multilevel linear and logistic mixed regressions were performed for continuous outcomes and binary outcomes, respectively, to assess the longitudinal associations between dietary patterns and cardiometabolic risk factors. Repeated measures (three timepoints) were reshaped in long-shape (three observations per individual) and analyzed in the mixed model, adjusting for wave. Polling areas (clusters) were set as the second level in the model, with participants at the first level. Three models were developed: Model 1, adjusted for age and sex; Model 2, additionally adjusted for marital status, education, and occupation; Model 3, additionally adjusted for leisure-time physical activity, alcohol consumption and tobacco use. To account for the potential effect of intervention and the interaction between intervention and time, a multiplicative term between the study group and timepoint was added in the model. All analyses were conducted using STATA16.0 (College Station, TX, USA: StataCorp LLC).

## 3. Results

### 3.1. Sample Characteristics

The characteristics of the study population at baseline have been reported previously [30], including that 47.2% of participants were women, and the mean age was 46.0 years (SD 7.5). At baseline, 69.6% of the study participants had central obesity, and the prevalence of prediabetes and metabolic syndrome was 72.1 and 37.3%, respectively. At 2 year follow-up, these numbers dropped to 61.5% for central obesity and 71.4% for prediabetes and increased to 38.5% for metabolic syndrome. There were 147 (16%) participants diagnosed with diabetes at 2 year follow-up [31].

### 3.2. Dietary Patterns

Two major dietary patterns were identified (Figure 2), explaining the total variance (of all the variables) of 19.2% (10.1 and 9.1% respectively). The same number of food groups were presented under each of the dietary patterns, but the order of the food groups presented under each pattern was different, based on the factor loading structure of each pattern. The “snack-fruit” pattern was characterized by food groups of fats, snacks, and fruits (as they have the highest factor loading on this pattern), and the “rice-meat-refined wheat” pattern was characterized by food groups of meat, rice, and refined wheat (as they have the highest factor loadings on this pattern).

### 3.3. Participants’ Characteristics by Dietary Patterns

Mean cumulative dietary pattern scores by baseline characteristics are presented in Table 1. The “rice-meat-refined” pattern score was significantly higher in younger groups (age ≤ 45 years), married people, males, people with a lower education level, those currently working, as well as among alcohol drinkers, tobacco users, and leisure inactive participants. Similar variables (sex, education, occupation, smoking, alcohol consumption, and leisure-time physical activity) were also significantly associated with the similar trends for the “snack-fruit” pattern but with smaller magnitude. The mean cumulative score for each of the dietary patterns did not differ between study groups and timepoints.

### 3.4. Participants’ Cardiometabolic Risks by Dietary Patterns

The mean cumulative dietary pattern scores by cardiometabolic risk factors at baseline are presented in Table 2. In brief, no differences in the mean cumulative score among the cardiometabolic risk factors were observed for the “snack-fruit” pattern. However, for the “rice-meat-refined wheat” pattern, a significantly higher score was found in hypertriglyceridemia (≥150 mg/dL (1.7 mmol/L, part of the definition for metabolic syndrome in the Methods) and marginally for low HDL.

Longitudinal associations between dietary patterns and cardiometabolic biomarkers are presented in Table 3. No associations were found between the “snack-fruit” pattern and the cardiometabolic risk factors, except for a positive association with triglycerides (mg/dL) (β = 6.76 95% CI 2.63–10.89). The “rice-meat-refined wheat” pattern was positively associated with Hb1Ac (%) (β = 0.04, 95%CI 0.01, 0.07).

As presented in Table 4, the “rice-meat-refined wheat” pattern was positively associated with central obesity (OR 1.16, 95% CI 1.01–1.34). No other associations were observed between the “rice-meat-refined” pattern and other cardiometabolic risk factors.

## 4. Discussion

In this study, we have identified two major dietary patterns (the “snack-fruit” pattern and the “rice-meat-refined wheat” pattern”) in a cohort of individuals at a high risk for diabetes in Kerala, India. We also have examined the longitudinal associations between these dietary patterns and a series of cardiometabolic risk factors, as well as metabolic syndrome. The “snack-fruit” pattern was significantly associated with increased triglycerides, whereas the “rice-meat-refined wheat pattern” was significantly associated with central obesity and elevated Hb1Ac.

Partly due to the huge differences in dietary intake across different Indian regions and states, dietary patterns identified in different studies are hard to compare [8]. However, there are some interesting similarities and dissimilarities in dietary patterns to be further understood. For example, the “snack-fruit” pattern identified in our study is similar to the “snack and fruit” pattern or “snacks and sweets” patten identified in South India previously. It contains both fruits (healthy) and snacks and sweets (unhealthy) [12,26]. Besides the food habits, other factors can be shared in the same region, such as socioeconomic status of households, as well as the local norms on food choice [32]. The “healthy” and “unhealthy” combined dietary patterns have been less observed in Western diets but are more commonly observed in India. Interestingly, despite the similarities with the aforementioned studies, “rice-meat-refined wheat” pattern identified in our study was different from the “pulse and rice” pattern [12] or the “lacto-vegetarian pattern” [26] detected in the same region. However, the common foods are rice/fermented rice, which are commonly eaten in South India (also often accompanied with meat). It is noted that although beef was loaded highly on the “rice-meat-refined wheat” pattern (indicating a high correlation between beef and this pattern), almost two thirds of the study population don’t consume beef, suggesting people that belonged to this pattern may mainly have other food groups featured on this pattern, such as rice, poultry, and refined wheat.

The “snack-fruit” pattern was found to be significantly associated with increased triglycerides in this study. This is different from another study that found that a similar “snacks and sweets” pattern was associated with central obesity but not lipids [12]. However, our “snack-fruit” pattern, characterized by coconut (high in saturated fats), fats, and oils, as well as sweets (high in sugar), may explain the association with triglycerides. We also found the “rice-meat-refined wheat” pattern was positively associated with central obesity. The studies looking at the associations between dietary patterns and obesity or central obesity have mixed results. The Chinese Nutrition and Health Survey (CHNS) has suggested that a “Traditional” pattern (high in rice, meat and vegetables) was protective for cardiometabolic risk factors, including obesity and central obesity [33]. However, this might be partly due to a diversity of food intake and the traditional Chinese cooking methods, which involve boiling and steaming. On the other hand, the “rice-meat-refined wheat” pattern found in the present study may highlight a high-fat intake [34], considering Indian cooking methods, as well as the high intake of fine carbohydrates (e.g., rice and refined wheat) [35]. This might also be partly due to the nutrition transition in India, which is shifting from diets high in cereals and fiber to diets high in sugar, fats, and animal foods, which are responsible for the increase in obesity and other cardiometabolic diseases [14]. In addition, the “rice-meat-refined wheat” pattern was only associated with central obesity but not obesity, which may be particularly seen in the South Asian population as extra body fat even with a normal BMI, as a thin–fat obesity phenotype is typical in the Indian population [36]. Studies have already suggested that effective interventions focused on central obesity or abdominal obesity in South Asian population is largely needed to reduce the elevated cardiometabolic risks [37].

Besides central obesity, the “rice-meat-refined wheat” pattern was also associated with Hb1Ac level, which has been rarely reported in India studies. This may be due to the high cost and availability of Hb1Ac tests in Indian settings. In a recent systematic review of randomized control studies on the effect of low glycemic index (GI) or dietary patterns on glycemic control suggested that low GI foods or dietary patterns reduced HbA1c when compared with higher GI foods or dietary patterns [38]. The “rice-meat-refined wheat” pattern identified in the present study is characterized by rice and refined wheat (high GI) and meat (e.g., beef and poultry). Given the low consumption of beef in the study population, the participants with this pattern may still consume rice (may accompany it with poultry) and refined wheat, mainly. Moreover, Hb1Ac is an important indicator of long-term glycemic control, with the ability to reflect the cumulative glycemic history of three months and being a reliable measure of chronic hyperglycemia, as well as long term risk diabetes outcomes [39]. In addition, although the association between the “rice-meat-refined wheat” pattern and newly developed diabetes at follow-up was not statistically significant, it seems to indicate a potential prediction of the “rice-meat-refined wheat” on future diabetes development. Such an association can be made stronger with longer follow-up, as the potential effect of the intervention (on diet) can be further attenuated. This may particularly alert the risk of potential diabetes and other cardiometabolic risks on this particularly dietary pattern, which is emerging in the nutrition transition among South Asian population, who are at higher risk of cardiometabolic diseases than other populations. The findings on the “rice-meat-refined wheat” pattern and central obesity, as well as HbA1c, are important for designing potential interventions in high-risk population of diabetes in the Indian setting.

One of the major strengths of our study is that we were able to conduct longitudinal analysis on the associations between dietary patterns and cardiometabolic risk factors among high-risk individuals for diabetes in an Indian setting. This provides better understanding of the complex associations between diet and disease, as most studies conducted in India have been cross-sectional in nature. Furthermore, we included a range of clinical biomarkers in our analyses, unlike previous studies in India which mostly relied on anthropometric and self-reported measures. Despite these strengths, there are some limitations. Firstly, the FFQ used in this study was not validated and the food items included were fewer than those analyzed in other studies, including those conducted in India. However, the FFQ was adapted from a previous study which was developed based on food items that are commonly consumed in Kerala [25]. Furthermore, the actual food items included into the calculations were almost doubled by including responses to “other food items” under each food group. Secondly, we only had data for a 2 year follow-up period, so the effect of dietary patterns may not be fully reflected in this short period in this population. Moreover, as the study population is from the K-DPP trial, the dietary patterns identified may be affected by the intervention itself. However, the mean cumulative dietary pattern scores did not differ between timepoints and study groups. Lastly, the findings may not be generalized to other states in India, given the differences in dietary intake across states in India.

## 5. Conclusions

In conclusion, our study findings show that there are two major dietary patterns (“snack-fruit” pattern and “rice-meat refined wheat” pattern) in a population who are at high risk for diabetes in Kerala, India. There is no clear line between “healthy” and “unhealthy” diets of these two dietary patterns, as each of them probably contribute to cardiometabolic risks in different ways. However, although elevated triglycerides tend to be associated with increased risks of cardiovascular diseases, it cannot serve as an independent marker for increased cardiovascular events, particularly in people with prediabetes and diabetes [40]. Therefore, attention should be given to the “rice-meat-refined wheat” pattern, given its likely effect on central obesity and Hb1Ac in Indian populations, who are at higher cardiometabolic risk than other populations. The relationship between diet and chronic disease risk is certainly complex, and studying this association in the Indian context is even more complex as the diet is far more diverse from most Western countries. These findings highlight the importance of nutrition in cardiometabolic risk reduction and early cardiovascular disease prevention. They are likely to inform the designing of dietary interventions for the prevention of cardiometabolic risk factors and diabetes, as well as early cardiovascular disease prevention in high-risk individuals in the Indian setting.

## Figures and Tables

**Figure 1 nutrients-14-00662-f001:**
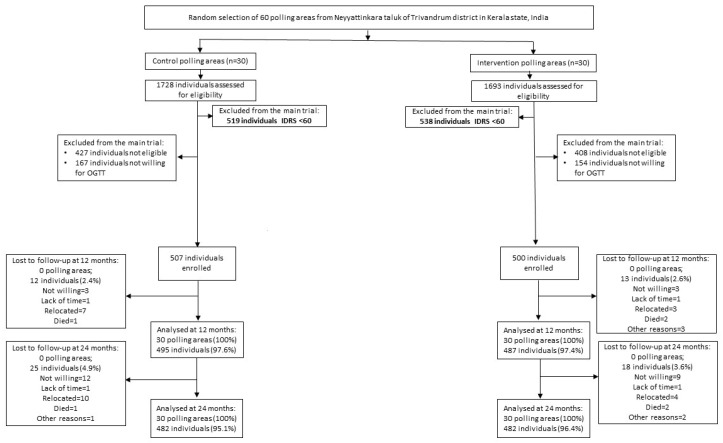
Flow chart for K-DPP participants across three time points. The “Lost to follow-up at 24 months” box is the cumulative loss from baseline to 24 months.

**Figure 2 nutrients-14-00662-f002:**
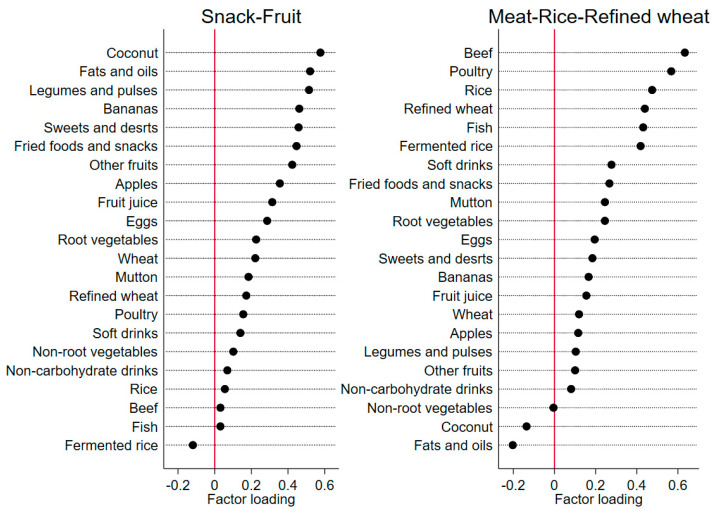
Dietary pattern loading by 22 food groups during three time points for the K-DPP study (n = 958, with non-missing dietary pattern across three timepoints).

**Table 1 nutrients-14-00662-t001:** The mean cumulative dietary pattern scores by baseline characteristics among K-DPP participants (*n* = 958).

	Snack-Fruit Pattern (Mean (SD))	*p*-Value	Rice-Meat-Refined Wheat Pattern (Mean (SD))	*p*-Value
Age		0.77		0.016
≤45 years (*n* = 463)	−0.0 (0.7)		0.0 (0.8)	
>45 years (*n* = 495)	−0.0 (0.7)		−0.1 (0.7)	
Sex		<0.001		<0.001
Male (*n* = 491)	0.1 (0.7)		0.3 (0.8)	
Female (*n* = 467)	−0.1 (0.6)		−0.4 (0.5)	
Marital status		0.22		<0.001
Married (*n* = 910)	−0.0 (0.7)		0.0 (0.7)	
Separated/divorced/widowed (*n* = 38)	0.1 (0.9)		−0.5 (0.7)	
Never married (*n* = 10)	0.3 (0.8)		−0.5 (0.4)	
Education		0.045		0.022
Up to primary school (*n* = 241)	−0.1 (0.7)		−0.0 (0.7)	
Secondary school (*n* = 564)	0.0 (0.7)		0.0 (0.7)	
Tertiary and above (*n* = 153)	0.0 (0.8)		−0.2 (0.6)	
Occupation		<0.001		<0.001
Skilled/unskilled (*n* = 683)	0.1 (0.7)		0.1 (0.8)	
Homemaker/unemployed/retired (*n* = 275)	−0.2 (0.6)		−0.4 (0.5)	
Leisure-time physical activity		<0.001		0.015
Leisure inactive (*n* = 762)	−0.1 (0.7)		−0.0 (0.7)	
Leisure active (*n* = 196)	0.2 (0.7)		0.1 (0.9)	
Alcohol use		0.001		<0.001
No (*n* = 762)	−0.0 (0.7)		−0.2 (0.6)	
Yes (*n* = 196)	0.1 (0.7)		0.5 (0.9)	
Tobacco use		<0.001		<0.001
No (*n* = 777)	−0.0 (0.7)		−0.1 (0.6)	
Yes (*n* = 181)	0.2 (0.8)		0.4 (0.9)	

**Table 2 nutrients-14-00662-t002:** Mean cumulative dietary pattern scores (across three years) by cardiometabolic risk factors at baseline (*n* = 985, with non-missing dietary data across three waves).

Cardiometabolic Risk Factors	Snack-Fruit Pattern (Mean (SD))	*p*-Value	Rice-Meat-Refined Wheat Pattern (Mean (SD))	*p*-Value
Obesity ^1^		0.39		0.59
No (*n* = 506)	0.0 (0.7)		−0.0 (0.7)	
Yes (*n* = 452)	−0.0 (0.7)		−0.0 (0.8)	
Central obesity ^2^		0.31		0.52
No (*n* = 289)	0.0 (0.8)		0.0 (0.7)	
Yes (*n* = 667)	−0.0 (0.7)		−0.0 (0.7)	
Hypertriglyceridemia ^3^		0.12		<0.001
No (*n* = 760)	−0.0 (0.7)		−0.1 (0.7)	
Yes (*n* = 198)	0.1 (0.7)		0.2 (0.8)	
Low HDL ^4^		0.16		0.055
No (*n* = 630)	0.0 (0.7)		0.0 (0.7)	
Yes (*n* = 328)	−0.1 (0.7)		−0.1 (0.7)	
Elevated blood pressure ^5^		0.57		0.20
No (*n* = 639)	0.0 (0.7)		−0.0 (0.7)	
Yes (*n* = 319)	−0.0 (0.7)		0.0 (0.8)	
Prediabetes ^6^		0.89		0.74
No (*n* = 271)	−0.0 (0.7)		−0.0 (0.8)	
Yes (*n* = 687)	−0.0 (0.7)		−0.0 (0.7)	
Metabolic syndrome ^7^		0.89		0.76
No (*n* = 602)	−0.0 (0.7)		−0.0 (0.7)	
Yes (*n* = 356)	−0.0 (0.7)		−0.0 (0.7)	

^1^ Obesity was defined as BMI ≥ 25 kg/m^2^ [22]; ^2^ Central obesity was defined as waist circumference ≥90 cm for males and ≥80 cm for females [23]; ^3^ Hypertriglyceridemia was defined as triglycerides ≥150 mg/dL (1.7 mmol/L); ^4^ Low HDL was defined as HDL <40 mg/dL (1 mmol/L) for males and <50 mg/dL (1.3 mmol/L) for females; ^5^ Elevated blood pressure was defined as systolic blood pressure ≥130 mmHg and /or diastolic blood pressure ≥ 85 mmHg or drug treatment for hypertension; ^6^ Prediabetes was defined according to the ADA: 5.6 ≤ Fasting glucose ≤ 6.9 or 7.8 ≤ 2-h glucose ≤ 11.0, or 5.7% ≤ HBA1c ≤ 6.4% [24]; ^7^ Metabolic syndrome was defined as central obesity + any two or more symptoms defined above from item 3–6 [23].

**Table 3 nutrients-14-00662-t003:** Longitudinal associations between dietary patterns and cardiometabolic biomarkers using multi-level mixed effects models ^1^.

	Snack-Fruit Pattern	Rice-Meat-Refined Wheat Pattern
Triglycerides (mg/dL)		
Model 1	7.44 (3.30, 11.58)	1.23 (−3.11, 5.58)
Model 2	7.59 (3.41, 11.78)	1.84 (−2.57, 6.25)
Model 3	6.76 (2.63, 10.89)	−1.34 (−5.75, 3.06)
HDL cholesterol (mg/dL)		
Model 1	−0.41 (−1.18, 0.36)	−0.09 (−0.90, 0.72)
Model 2	−0.59 (−1.37, 0.18)	0.08 (−0.74, 0.89)
Model 3	−0.55 (−1.32, 0.22)	−0.37 (−1.19, 0.45)
Systolic blood pressure (mmHg)		
Model 1	−0.80 (−1.75, 0.14)	0.08 (−0.92, 1.08)
Model 2	−0.90 (−1.85, 0.05)	0.34 (−0.67, 1.35)
Model 3	−0.87 (−1.82, 0.07)	−0.05 (−1.07, 0.97)
Diastolic blood pressure (mmHg)		
Model 1	−0.45 (−1.10, 0.19)	−0.54 (−1.19, 0.11)
Model 2	−0.33 (−1.02, 0.35)	−0.18 (−0.88, 0.51)
Fasting glucose (mmol/L)		
Model 1	0.01 (−0.03, 0.05)	0.03 (−0.02, 0.07)
Model 2	−0.00 (−0.04, 0.04)	0.04 (−0.01, 0.08)
Model 3	0.00 (−0.04, 0.04)	0.04 (−0.01, 0.08)
Two-hour glucose (mmol/L)		
Model 1	−0.05 (−0.16, 0.06)	0.02 (−0.09, 0.14)
Model 2	−0.05 (−0.16, 0.06)	0.04 (−0.08, 0.16)
Model 3	−0.04 (−0.15, 0.07)	0.04 (−0.08, 0.16)
Hb1Ac (%)		
Model 1	0.01 (−0.02, 0.03)	0.04 (0.01, 0.07)
Model 2	0.00 (−0.03, 0.03)	0.05 (0.01, 0.08)
Model 3	0.00 (−0.03, 0.03)	0.04 (0.01, 0.07)

^1^ Results are presented as beta coefficients with 95% CI. Model 1: age, sex. Model 2: Model 1 + marriage status, education, occupation. Model 3: Model 2 + leisure PA, alcohol consumption and smoking. In addition, the model was adjusted for wave, study group, and the interaction between timepoint and study group.

**Table 4 nutrients-14-00662-t004:** Longitudinal associations between dietary patterns and binary cardiometabolic risk factors using multi-level mixed-effects models *.

	Snack-Fruit Pattern	Rice-Meat-Refined Wheat Pattern
Obesity		
Model 1	0.97 (0.86, 1.10)	1.01 (0.89, 1.15)
Model 2	0.98 (0.87, 1.12)	1.02 (0.89, 1.16)
Model 3	0.97 (0.85, 1.10)	1.00 (0.87, 1.14)
Central obesity		
Model 1	1.03 (0.90, 1.17)	1.18 (1.03, 1.35)
Model 2	1.03 (0.90, 1.17)	1.19 (1.03, 1.36)
Model 3	1.02 (0.90, 1.16)	1.16 (1.01, 1.34)
Hypertriglyceridemia		
Model 1	1.08 (0.94, 1.24)	1.00 (0.87, 1.16)
Model 2	1.08 (0.94, 1.25)	1.03 (0.89, 1.18)
Model 3	1.05 (0.91, 1.22)	0.96 (0.82, 1.11)
Low HDL		
Model 1	0.96 (0.85, 1.09)	0.95 (0.84, 1.08)
Model 2	0.98 (0.87, 1.11)	0.94 (0.82, 1.07)
Model 3	0.97 (0.86, 1.10)	0.95 (0.83, 1.09)
Raised blood pressure		
Model 1	0.89 (0.78, 1.01)	1.05 (0.92, 1.20)
Model 2	0.90 (0.79, 1.02)	1.07 (0.94, 1.22)
Model 3	0.90 (0.79, 1.03)	1.03 (0.89, 1.18)
Diabetes		
Model 1	1.04 (0.80, 1.35)	1.20 (0.93, 1.56)
Model 2	0.99 (0.76, 1.29)	1.28 (0.98, 1.66)
Model 3	0.99 (0.76, 1.29)	1.28 (0.98, 1.67)
Metabolic syndrome		
Model 1	0.99 (0.87, 1.13)	1.14 (0.99, 1.30)
Model 2	1.00 (0.87, 1.13)	1.14 (1.00, 1.31)
Model 3	0.99 (0.87, 1.13)	1.11 (0.97, 1.28)

* Results are presented as odds ratio and 95% CI. Instead of reporting prediabetes, which takes 72% of the study population at baseline, and didn’t change too much along the waves, diabetes (about 15% developed diabetes at year 2) is reported here, using the dietary cumulative mean score to predict diabetes at year 2. Model 1: age, sex. Model 2: Model 1 + marriage status, education, occupation. Model 3: Model 2 + leisure PA, alcohol consumption, and smoking. In addition, the model was adjusted for wave, study arm, and the interaction between wave and study arm for all the other outcomes in the table, but not for diabetes. For diabetes (developed at year 2), only study arm was adjusted.

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
