# Peer review of "Associations between Dietary Patterns and Cardiometabolic Risk Factors—A Longitudinal Analysis among High-Risk Individuals for Diabetes in Kerala, India"

_nutrients, 2022, doi:10.3390/nu14030662_

Round 1
Reviewer 1 Report
To the authors
This manuscript is reporting association between cardiometabolic risk factors and the structure of India diet. The authors found two major structural elements and their association with hypertriglyceridemia, central obesity, and HbA1c level. This manuscript may contribute for the readers to understand the role of diet in metabolic syndrome.
There are some comments.
- Methods: I cannot understand how to perform the longitudinal analysis. Please describe the statistical methods in detail.
- Results: I think that a PCA plot helps readers to understand their results.
- In 3.4., the authors should describe “at baseline” in the main text.
- It was confusing the words “dietary pattern” used in this manuscript. I think that this dietary pattern indicates a component or axis in PCA. If so, I recommend that the authors should change the words to other easily understandable words.
- Page 4, line 145: “kg/m2” should read “kg/m2”.
- Page 4, line 151: Usually, the words “blood sugar” should be expressed as “blood glucose” in an academic article.
Reviewer 2 Report
The manuscript is interesting and original, and is worthy of publication. However, I recommend emphasizing in the conclusions the importance of early cardiovascular prevention and how nutrition represents one of the pillars together with the prevention of cardiometabolic diseases , especially evident in patients at higher risk such Metabolic Syndrome, erectile dysfunction etc.
- D'Ascenzi F, et al. When should cardiovascular prevention begin? The importance of antenatal, perinatal and primordial prevention. Eur J Prev Cardiol. 2019 Dec 16:2047487319893832.
- Di Francesco S, et al. Mediterranean diet and erectile dysfunction: a current perspective. Cent European J Urol. 2017;70(2):185-187.
- Li D et al. Lifetime risk of cardiovascular disease and life expectancy with and without cardiovascular disease according to changes in metabolic syndrome status. Nutr Metab Cardiovasc Dis. 2022;32(2):373-381.
